# Study of Heat Flow at Substrate during Sputtering of Copper–Titanium Sandwich Target

**DOI:** 10.3390/ma17143599

**Published:** 2024-07-21

**Authors:** Viktor I. Shapovalov, Daniil S. Sharkovskii

**Affiliations:** Department of Physical Electronics and Technology, St. Petersburg Electrotechnical University “LETI”, Prof. Popov str., 5F, St. Petersburg 197022, Russia

**Keywords:** magnetron sputtering, sandwich target, heat flow, thermocouple sensor, heating the substrate, Fourier equation

## Abstract

The purpose of this work is to study the kinetics of the heat flow heating the substrate, which is generated by a two-layer sandwich magnetron target when sputtered in argon. Its novelty resides in the application of the COMSOL Multiphysics to study the kinetics of thermal processes during sputtering of a target of the new type. The analysis was performed for a sandwich target with internal copper and external titanium plates when the discharge power varied in the range of 400–1200 W. The heating of the external target plate is described by a two-dimensional homogeneous Fourier equation. The solution to the equation reveals how the kinetics of the external plate’s surface temperature distribution depends on the discharge power. To study the heat flow heating the substrate, the external plate is presented in the form of an additive set of small-sized surface heat sources. Previously unknown features of the thermal process are established. It is shown that numerical modeling adequately describes the experimental results.

## 1. Introduction

In modern technology, films and coatings play an important role. If one pays attention, for example, to any engineering industry, aircraft and shipbuilding, and construction of buildings and structures, then there will definitely be a place where film materials with a thickness from fractions to hundreds of micrometers are used to modify the surfaces of products.

Inorganic films and coatings in the form of binary alloys or binary solid solutions of simple compounds (oxides, nitrides, etc.) of various metals attract much attention around the world. Currently, many leading research centers synthesize these coatings. For example, TiAlN nitride is one of the most popular materials [1,2]. In addition, researchers are also highly interested in TiTaO [3], TiAlO [4], TiCrO [5], and MoAlO [6], or nitrides TiCrN [7], MoCrN [8], etc.

To deposit such films, two magnetrons with cold targets [1], a magnetron with a combined cold target [8], or a magnetron with a single pressed target [9,10] are used. Each of these methods has advantages and disadvantages.

At the same time, the magnetrons with targets that can be heated up to the melting temperature and above are also studied [11,12,13,14,15]. The hot target mode is ensured by removing heat from the target through a gap up to 1 mm wide and along the fastening elements. While developing the magnetron sputtering technique, we proposed a magnetron equipped with a sputtering unit, which we called a sandwich target [16,17,18]. A sandwich target, which can be used for depositing films of binary alloys or binary solid solutions of simple metal compounds, contains two parallel plates fixed with a small gap (1–2 mm) on one axis. The internal plate operates in a cold mode. The external one is hot and has cut-outs through which the internal plate is sputtered. The studies of sandwich target magnetrons are in the early stages.

Substrate heating by magnetron sputtering has always been of great interest [19,20,21,22,23,24,25,26,27,28,29,30]. One of the first works [19] presented the results of measuring the kinetics of substrate heating during dc magnetron sputtering of some metals in an argon environment. It was found that heating of the substrate is caused by the kinetic energy of sputtered atoms; energy released during phase transitions (condensation and crystallization of the deposited material); discharge radiation and kinetic energy of electrons from the discharge [19,23]. Attention to this problem still exists [31,32]. All these works were performed for magnetrons with effectively cooled targets. Most of the works were devoted to experimental research. Some publications presented simulation results [20]. Our publications are devoted to the study of thermal processes during magnetron sputtering of targets operating in hot mode [17,33]. Using the COMSOL Multiphysics software, processes were studied in steady operating modes of magnetrons with two-layer targets. In this case, the heating of the target elements, which are heat sources, was assessed by the effective temperature. The main difficulty in calculating this quantity, as will be shown further, lies in the uncertainty in the choice of the emitting region on their surfaces.

This work extends the research described in [17]. Its goal was to study the kinetics of the heat flow heating the substrate, which is generated by a two-layer sandwich magnetron target when sputtered in argon. At the same time, a more correct method was used to estimate the temperature of the external plate.

The studies were performed for a sandwich target containing internal copper and external titanium plates. Such a target is intended for deposition of films of the Cu–Ti alloy, its intermetallic compounds CuTi, CuTi_2_, or solid solutions of the corresponding oxides, nitrides, etc. Their good mechanical strength, low weight, and good oxidation resistance, especially at increased temperatures, attract the interest of researchers [34]. They are used in microelectronics due to their high conductivity [35]. For medical purposes, they are applied mainly due to their mechanical strength, biocompatibility, and antibacterial properties [36].

## 2. Experimental

In this work, a sandwich target mounted on a flat cylindrical balanced dc magnetron with a diameter of 130 mm was studied. The target (see Figure 1) consisted of a 4 mm thick internal copper plate (*1*) cooled by water, on which a 1 mm thick external titanium plate (*2*) operating in hot mode was fixed with a gap of 1 mm, and fastening rings (*3*). In the experimentally established ring-shaped sputtering region (*4*) of the external plate with an area of 36.5 cm^2^ and the radii of the internal and external regions of 19 and 39 mm, respectively, eight cut-outs (*5*) with a diameter of 12 mm were made. This choice provided a 1:3 ratio between the areas of the internal and external plates that were sputtered. To experimentally study the thermal processes occurring on the substrate, a thermocouple sensor with a sensing element in the form of a copper disk with an area of 100 mm^2^ was used. Thermal measurements were carried out at a discharge current of 1–3 A and an argon pressure of 2–6 mTorr with a relative error of ±10%.

To perform the calculations, it was necessary to establish the possible range of power released on the target during sputtering. This was conducted according to the current–voltage characteristics (I–V characteristics) of the discharge shown in Figure 2. The I–V characteristics are measured in the range of currents at which the magnetron is stable and the external plate does not reach the melting temperature. It is typical that the current–voltage characteristics of a magnetron with a sandwich target contain a maximum caused by the occurrence of significant thermionic emission at a current of more than 3 A. Therefore, the dependence of the power on the discharge current (see Figure 3) is nonlinear.

Figure 3 shows that the power released on the target in the working range of currents varies in the range of approximately 400–1200 W, which was adopted for the study. In addition, all calculations were performed for a pressure of 4 mTorr. Although this choice is arbitrary, Figure 3 shows that the results will be qualitatively similar for other pressures as well. An increase or decrease in power with a change in pressure will only lead to a corresponding change in heat flows in the sputtering system. For a similar reason, the total relative area of the cut-outs in the external plate was chosen to be 0.25. Changing it will entail a redistribution of the power released on the plates. In this case, heating of the external plate will increase or decrease, but the process will change little qualitatively.

The Heat Transfer Module of the COMSOL Multiphysics was used to perform a numerical study. The peculiarities of the geometric model of the vacuum chamber–magnetron target system can be found in [33]. The boundary conditions of the problem are also given there. T(x,y,0)=293 K is taken as the initial condition.

Further, when developing the technique, we will illustrate it with the results of calculations for 1000 W power released on the target. In conclusion, using the developed technique, we will perform an analysis of thermal processes for other power values from the range set in this section.

## 3. Results and Discussion

The heating kinetics of a sandwich target were studied by changing the distribution of the external plate surface temperature. For this purpose, a two-dimensional homogeneous Fourier equation was solved:(1)$$\frac{dT(x,y,t)}{dt}-a^{2}\Delta T(x,y,t)=0$$
where *x* and *y* are the coordinates on the surface of the plate; *a* (m^2^/s) is the thermal diffusivity of the external plate.

For example, Figure 4 shows the results of solving Equation (1) as parts of typical instantaneous distributions of temperature *T*(*x*, *y*, *t*) at *x* > 0 *y* = 0 in the axial section passing between the cut-outs (see Figure 1b). It is obvious that in sections that include cut-outs, the temperature in these intervals will decrease abruptly to values corresponding to the temperature of the internal plate. A steady state occurs at *t* > 500 s. A characteristic feature of the distributions in Figure 4 is the maxima located closer to the center outside the sputtering area of the ring-shaped plate. Figure 5 shows kinetic curves for four characteristic points in the cross-section of the plate (*x* = 0 is the center; *x*_Tmax_ is the point of maximum temperature; *x*_Tmax_/^2^ is the point corresponding to half of the maximum temperature; *x* = 65 mm is the point at the edge of the plate). The curves in Figure 5 are plotted according to the distributions shown in Figure 4. As Figure 5 shows, the process of increasing temperature at all points of the plate is exponential.

There is no reason to believe that, right up to melting, the general instantaneous distribution of the external plate surface temperature depends on the power released on the target. Such a dependence is possible only for the main parameters of this distribution, which include the temperature in a steady state calculated at each point of the external plate surface, and the time constant to reach it.

The source of heat flow during sputtering of a sandwich target is its external plate cooled by radiation and by fastening elements. The result of solving Equation (1) makes it possible to determine the kinetics of the heat flow *Q*_rad_. This flow can be related to the effective temperature *T*_eff_:(2)$$T_{\mathrm{eff}}\left(t\right)=\frac{1}{F_{\mathrm{rad}}}\iint_{F_{\mathrm{rad}}}T(x,y,t)dxdy$$
where $F_{\mathrm{rad}}$ (m^2^) is the area of the radiating surface of the upper plate. The relation between *Q*_rad_ (W) and $T_{\mathrm{eff}}$ (K) is set by the Stefan–Boltzmann law:(3)$$Q_{\mathrm{rad}}\left(t\right)=\sigma (\varepsilon_{t}T_{\mathrm{eff}}^{4}\left(t\right)-\varepsilon_{\mathrm{rec}}T_{\mathrm{rec}}^{4})F_{\mathrm{rad}},$$
where σ = 5.67 × 10^−8^ W∙m^−2^∙K^−4^ is the Stefan–Boltzmann constant; ε_t_ is the target emissivity; ε_rec_ and *T*_rec_ are the emissivity and temperature of the radiation receiver. In (3), the values of *T*_eff_ and *T*_rec_ should be substituted in kelvins; in other cases, the temperature will be expressed in degrees Celsius.

The technique for determining the kinetics of the heat flow generated by the sandwich target, based on the direct application of (3) with integral (2), is called integral. The main difficulty of its application resides in the uncertainty of the selection of the radiating region on the surface of the external plate. Figure 6 shows three variants of such regions as an example. Two of them are bounded by dashed circles. The third option is the total area of the plate.

Figure 7 presents the calculated kinetic curves reflecting the change in the effective temperature of the selected radiating regions of the external plate. Figure 7 shows that in the steady-state mode, the effective temperature of the external plate varying from 870 to 1750 K depends significantly on the selection of the radiating region. This means, based on (3), that the results of calculating the heat flow radiated by the plate will have a similar variation.

Let us consider a different technique, which we will call differential. In this case, the surface of the plate is represented by a series of concentric figures. The central region is formed by a circle at *x* = 0 with radius *r*_1_. All subsequent ones will be concentric rings with internal and external radii *r_i_* and *r*_i+1_ for *i* = 1, 2, …, *N* − 1, respectively. Let us assume that the temperature distribution on the surface of each ring is constant *T**(*x_i_*) and can be calculated as an average value:(4)$$T^{*}\left(x_{i}\right)=\frac{T\left(r_{i+1}\right)+T\left(r_{i}\right)}{2}$$
where $x_{i}=\left(r_{i+1}+r_{i}\right)/2$ is the radius of the central circle of the *i*-th ring; *T*(*r_i_*_+1_) and *T*(*r_i_*) are the calculated values of temperature *T*(*x*) at its boundaries. To find the values of *r_i_* and *T**(*x_i_*) *i* = 1, 2, …, *N* − 1, a stepwise approximation of the dependence *T*(*x*) was performed for each value of *t*. During the calculations, it was taken into account that *r*_i+1_ ≤ 65 mm.

For a central circle with radius *r*_1_, expression (4) takes the form
(5)$$T^{*}\left(0\right)=\frac{T\left(0\right)+T\left(r_{1}\right)}{2}$$
where *T*(0) is the calculated temperature value at the center of the plate (see Figure 4 at *x* = 0).

Let us further develop the technique for studying heat flow using the proposed model. Let us keep in mind that the ultimate goal is to study the kinetics of the heat flow incident on the surface of a remote substrate, which in this work is the sensitive element of a thermocouple sensor with an area of 100 mm^2^.

An example of the result of stepwise approximation taking into account (4) and (5) is shown in Figure 8. The approximation was performed with a uniform step of 1 mm, which ensured a relative error of less than 2%. Each fragment of the external plate, in accordance with (3), emits a heat flow:(6)$$Q_{\mathrm{rad}i}\left(t\right)=\sigma (\varepsilon_{t}T^{*4}\left(x_{i},t\right)-\varepsilon_{\mathrm{rec}}T_{\mathrm{rec}}^{4})\Delta F_{i},$$
where Δ*F_i_ i* = 0, 1, 2, …, *N* − 1 are the areas of the concentric figures of the external plate (*i* = 0 corresponds to the central circle).

Figure 9 shows, as an example, part of the kinetic curves of the power emitted by fragments of the external plate. When performing the calculations, the wall of the vacuum chamber was taken as the radiation receiver at values ε_t_ = ε_rec_ = 0.1937 [37]. Since the area of the wall of the vacuum chamber significantly exceeds the area of the sensitive element of the thermal sensor, the wall with temperature of 293 K is the main receiver of radiation from the heated target. Then, we can assume that the inflection points on all dependencies shown in Figure 9 are due to the fact that with increasing temperature of the external plate, the influence of the vacuum chamber wall, having a constant temperature, on the energy exchange decreases.

When performing calculations, it was taken into account that some of the rings contain fragments of cut-outs through which the internal plate is sputtered. The radiating areas of these rings Δ*F_i_ i* = 0, 1, 2,…, *N* − 1 (see Equation (6)) were reduced accordingly.

Figure 10 shows the kinetics of the total thermal radiation of the target.
(7)$$Q_{\Sigma }\left(t\right)=\sum_{i=0}^{N-1}Q_{\mathrm{rad}i}\left(t\right).$$

As Figure 10 shows, in all cases, the radiated power in the steady mode does not exceed 40% of that released on the target.

On a remote surface, taking into account (6), the external plate creates a flow:(8)$$Q_{\mathrm{rad}}\left(t\right)=\sum_{i=0}^{N-1}\varphi_{i}Q_{\mathrm{rad}i}\left(t\right),$$
where φ*_i_* is the angle of visibility or angular coefficient of emissivity, defined as the fraction of the heat flow incident on a given surface from the total heat flow emitted by the source. Let us recall that the radiation receiver is a disk with an area of 100 mm^2^, which is located on the same axis with the external plate at a distance of 110 mm. Calculations were performed using techniques from [38].

Figure 11 shows diagrams of interaction with the sensing element of the thermocouple sensor of two typical radiating fragments of the target (lower figures in Figure 11), explaining the technique for calculating the angle of visibility. For the disk in Figure 11, a calculation is made using the formula
(9)$$\varphi_{0}=\frac{1}{2}\left(Z_{0}-\sqrt{Z_{0}^{2}-4R^{2}H_{0}^{2}}\right),$$
where $R=\frac{r_{sens}}{h},H_{0}=\frac{h}{r_{0}},Z_{0}=1+\left(1+R^{2}\right)H_{0}^{2}$

For the ring in Figure 11b, the following expression is used:(10)$$\varphi_{i}=\left(1+\frac{S_{i}}{S_{i+1}}\right)\varphi_{i+1,sens}-\frac{S_{i}}{S_{i+1}}\varphi_{i+1,sens},i=1,\ldots ,N-1,$$
where $S_{i}=\pi r_{i}^{2}$ is the area of the internal region of the ring with a radius *r_i_*; $S_{i+1}=\pi r_{i+1}^{2}$ is the disk area with a radius *r_i_*_+1_; $\varphi_{i+1,sens}=\frac{1}{2}\left(Z_{i}-\sqrt{Z_{i}^{2}-4R^{2}H_{i}^{2}}\right)$; $R=\frac{r_{sens}}{h},H_{i}=\frac{h}{r_{i}},Z_{i}=1+\left(1+R^{2}\right)H_{i}^{2}$.

The calculation results using Formulas (9) and (10) are shown in Figure 12, and it is clear that at *x* = 0 the viewing angle has a maximum and decreases towards the edge of the target. Some of the calculation results are summarized in Table 1.

Figure 13 shows the final result in the form of the kinetic dependence of the total flow falling on the surface of the thermal sensor’s sensing element in the steady state of the target. For the calculation, Formula (8) was used.

Checking the adequacy of the result in Figure 13 for the steady mode was carried out by comparison with the results of measuring the heating kinetics of the sensing element of the thermal sensor. Figure 14 shows typical experimental results obtained for the sandwich target described above at an argon pressure of 4 mTorr. The temperature of the sensing element in the steady mode based on Figure 14 changes, as shown in Figure 15.

Using the technique from [15], according to the curves in Figure 14, the heat flows heating the sensing element were determined (see Figure 16). The basis of the calculation was the kinetic equation for heating the sensing element [15]:
(11)$$\frac{dT}{dt}\left(T\right)=A-BT^{4}-CT,$$
where the first term *A* (°C/s) is proportional to the power released on the element. The other two terms describe heat removal from the element. Parameters *B* (1/((°C)^3^ × s)) and *C* (1/s) are proportional to the emissivity of the substrate, the thermal conductivity of the gas, and the structural elements of the substrate holder, respectively. At temperatures < 300 °C (see Figure 15), the radiation of the sensitive element can be neglected; therefore, calculations using (11) were performed at *B* = 0. The calculation results in the form of dependencies (11) are given in Figure 16. The values of parameter *A* are summarized in Table 2.

The last line of Table 2 contains the values of the power released on the sensitive element, obtained by the following formula:$$Q_{sens}=mcA,$$
where *m* and *c* are the mass (kg) and specific heat (J/(kg × °C) of the element, respectively. In this work, the product *mc* = 0.343 J/°C.

In Figure 17, dependence *2* was obtained using simulation. It does not fall into the area limited by the dashed lines. This area was built based on the results of the experiment from the last line of Table 2, taking into account a measurement error of ±10%. Note that the heat flow heating the sensitive element of the sensor at the highest discharge power of 1200 W (corresponding to a discharge current of 3 A) does not exceed 1 W. The difference between experiment and calculation observed in Figure 17 means that the heating of the sensitive element of the sensor occurs not only due to the thermal radiation of the target; there are additional heat flows, and the most significant of them are associated with the kinetic energy and heat of condensation of sputtered atoms, as well as discharge radiation [19].

Let us estimate the possible heat flows caused by these phenomena using data from [19]. Taking into account the characteristics of the sandwich target under study, the heat flow that heats the sensitive element of the thermocouple sensor, which is not related to the external target heating, will be given by expression
(12)$$Q_{\mathrm{sput}}=\left(\frac{0.25I}{e(1+\gamma_{\mathrm{Cu}}))}S_{\mathrm{Cu}}E_{\mathrm{Cu}1}+\frac{0.75I}{e(1+\gamma_{\mathrm{Ti}}))}S_{\mathrm{Ti}}E_{\mathrm{Ti}1}\right)\varphi ,$$
where *S*_Cu_ = 2.6 [39] and *S*_Ti_ = 0.7 [39] are the sputtering yields; γ_Cu_ = 0.079 [40] and γ_Ti_ = 0.11 [40] are ion-induced electron emission yields; *E*_Cu1_ ≈ 1.9×10^−18^ J/atom [19] and *E*_Ti1_ ≈ 3.5 × 10^−18^ J/atom [19] are the total energies per atom, including the kinetic energy of sputtered atoms and the heat of their condensation, as well as the radiation energy of the discharge; φ is the viewing angle of the sensing element by the sputtered target area. The subscripts Cu and Ti in (12) mean that this parameter refers to copper or titanium, respectively. The angle φ = 0.00228 in (12) was calculated using a scheme, where the rings on the surface of the external plate with *r*_1_ = 19 mm and *r*_2_ = 39 mm were taken as the heat flow emission region *Q*_sput_ (see Figure 11b). 

Expression (12) is written under the assumption of additivity of particle flows sputtered from the internal (first term) and external (second term) plates. Using factors 0.25 and 0.75 in (12), the redistribution of the discharge current between the plates is taken into account when the diameter of the cut-outs in the external plate is 12 mm. The influence of the discharge current on the additional heat flow estimated using (12) is shown in Table 3.

The total heat flow *Q_sens_* + *Q*_sput_ heating the substrate is shown in Figure 17 by dependence *3*. This result allows us to consider the thermal calculation performed in this work to be adequate.

## 4. Conclusions

In this work, we carried out a study of the kinetics of the heat flow heating the substrate, which is generated by a two-layer copper–titanium sandwich target of a magnetron when sputtered in argon at the discharge power in the range of 400–1200 W. Previously unknown features of this thermal process were found.

A characteristic feature of the instantaneous temperature distributions on the surface of the external plate is the maxima located closer to the center outside the sputtering region of the ring-shaped plate. The plate reaches a steady thermal mode in approximately 500 s. In this case, the kinetic curves of the radiated heat flow of the external plate contain inflection points, which are associated with the fact that with increasing temperature of the external plate, the influence of the wall of the vacuum chamber, which has a constant temperature, on the energy exchange decreases. The total heat flow heating the sensing element of the sensor at the highest discharge power of 1200 W did not exceed 1 W. It was shown that the numerical modeling adequately describes the results of the experiment.

## Figures and Tables

**Figure 1 materials-17-03599-f001:**
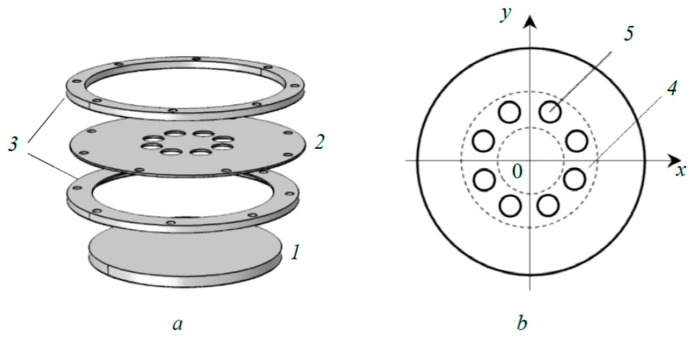
3D image of the sandwich target (**a**) and view of the external plate (**b**): *1*—internal plate; *2*—external plate; *3*—fastening rings; *4*—sputtering region; *5*—cut-out.

**Figure 2 materials-17-03599-f002:**
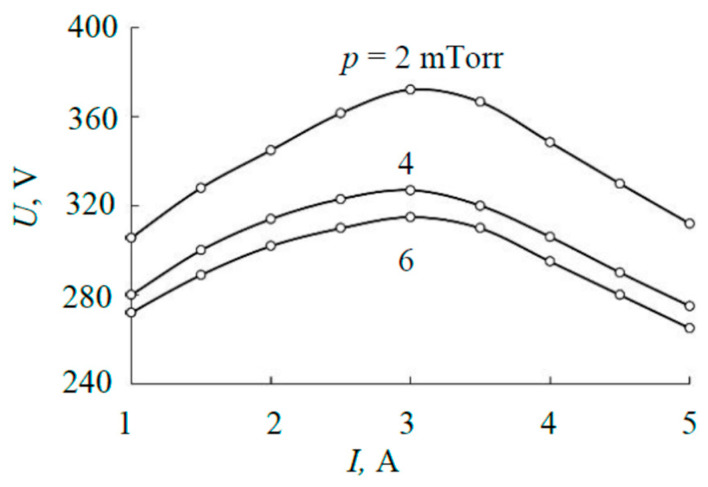
I–V characteristics of discharge of magnetron with the sandwich target.

**Figure 3 materials-17-03599-f003:**
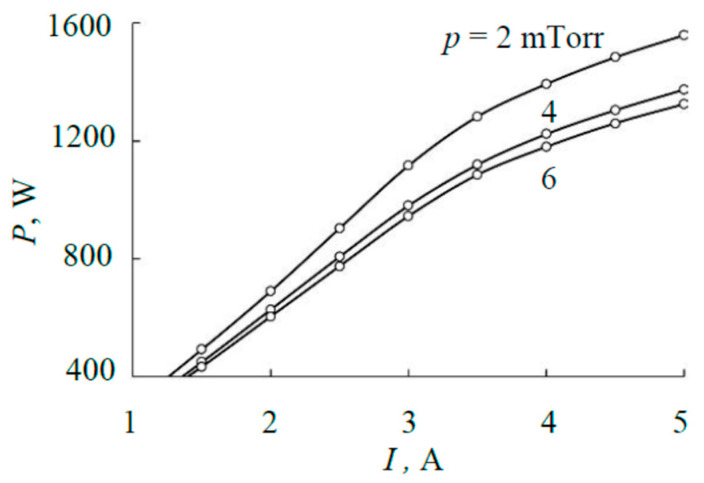
Power released on the target.

**Figure 4 materials-17-03599-f004:**
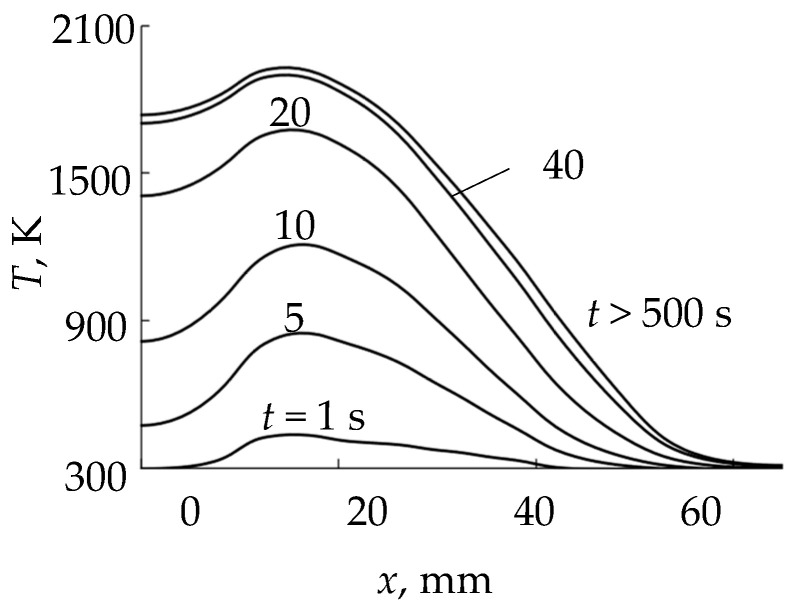
Distributions of the external plate surface temperature at different moments of time at power of 1000 W.

**Figure 5 materials-17-03599-f005:**
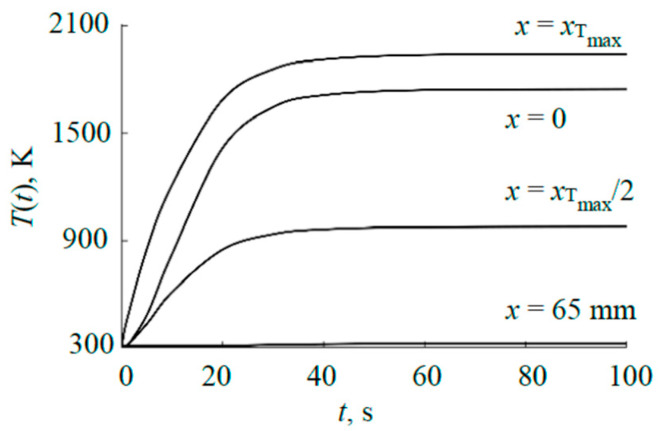
Kinetic heating curves of the external plate at characteristic points at power of 1000 W.

**Figure 6 materials-17-03599-f006:**
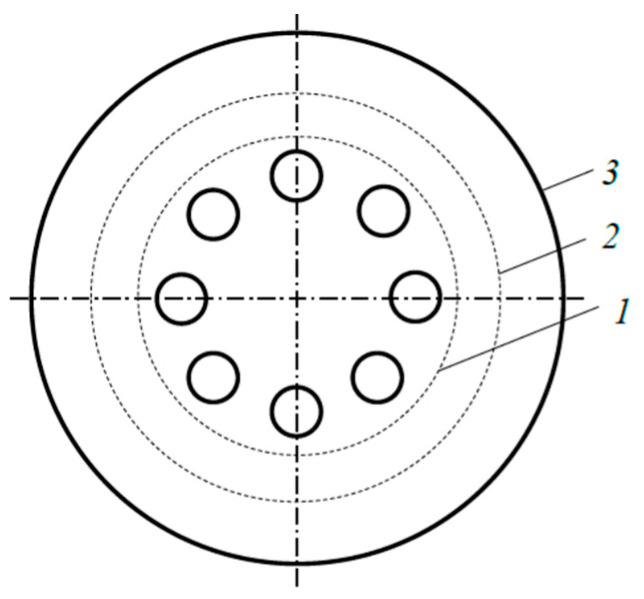
Radiating regions on the external plate with diameter *d*_rad_ (mm): *1*—78; *2*—100; *3*—130.

**Figure 7 materials-17-03599-f007:**
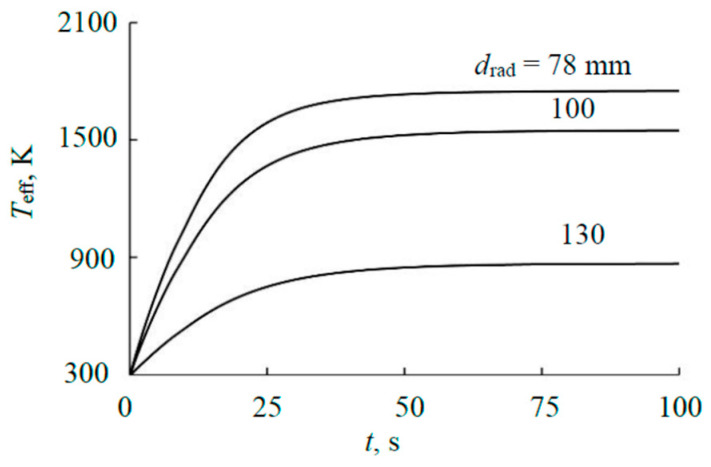
Kinetics of the external plate effective temperature at 1000 W and diameter *d*_rad_ of the radiating region.

**Figure 8 materials-17-03599-f008:**
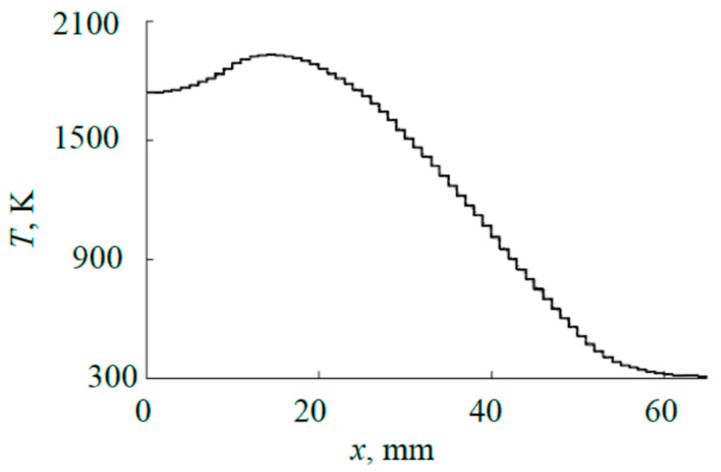
Approximation of external plate surface temperature distribution for steady mode at power of 1000 W.

**Figure 9 materials-17-03599-f009:**
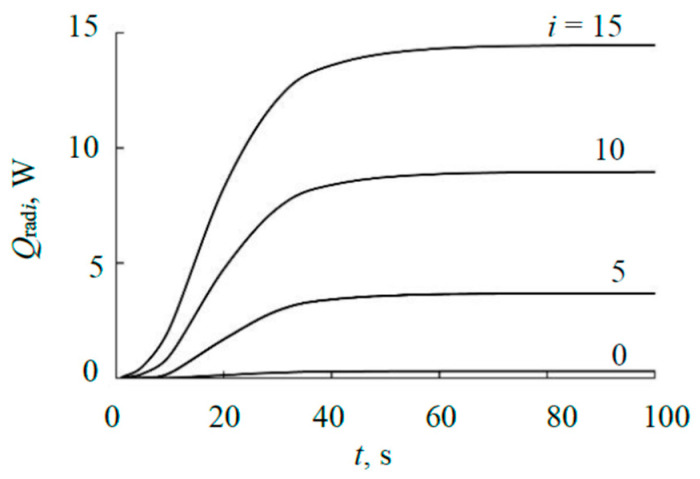
Kinetic curves of power emitted by *i*-th fragments of the external plate at power of 1000 W.

**Figure 10 materials-17-03599-f010:**
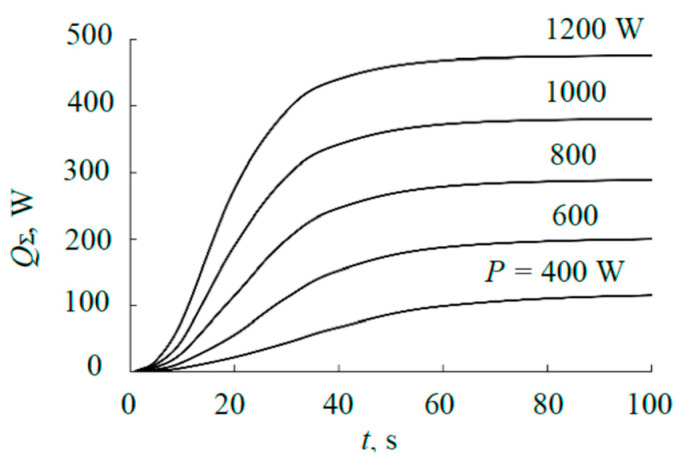
Kinetics of the total heat flow that radiates from the external plate of a sandwich target at a given power level.

**Figure 11 materials-17-03599-f011:**
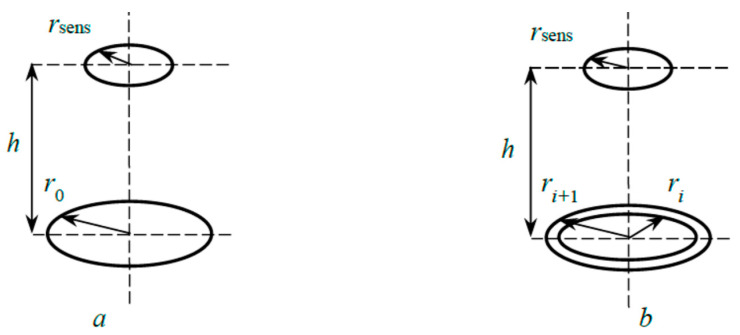
Schematic for calculation of visibility angles for target fragments: disk (**a**) and ring (**b**).

**Figure 12 materials-17-03599-f012:**
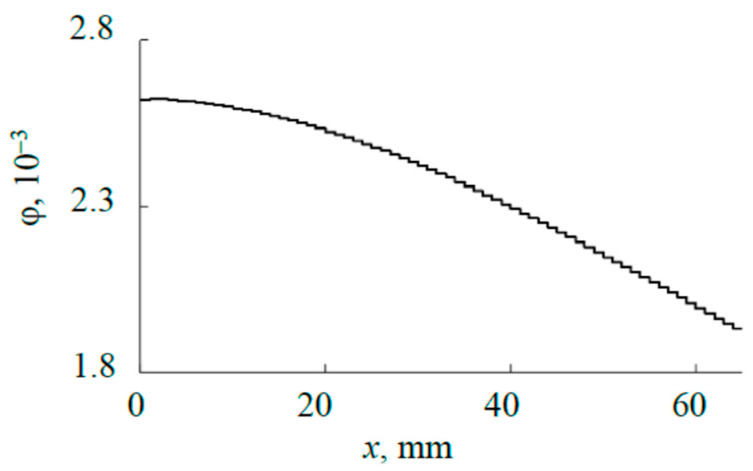
Change of the angular emissivity along the target surface.

**Figure 13 materials-17-03599-f013:**
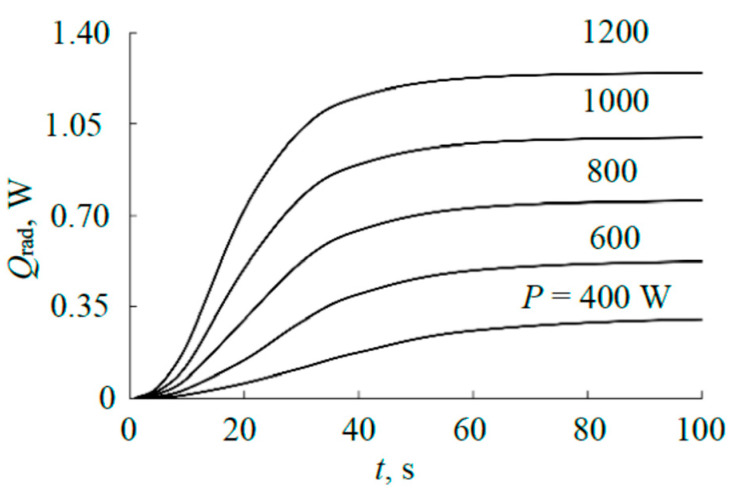
Kinetics of the total heat flow that reaches the substrate at a given power released on the target.

**Figure 14 materials-17-03599-f014:**
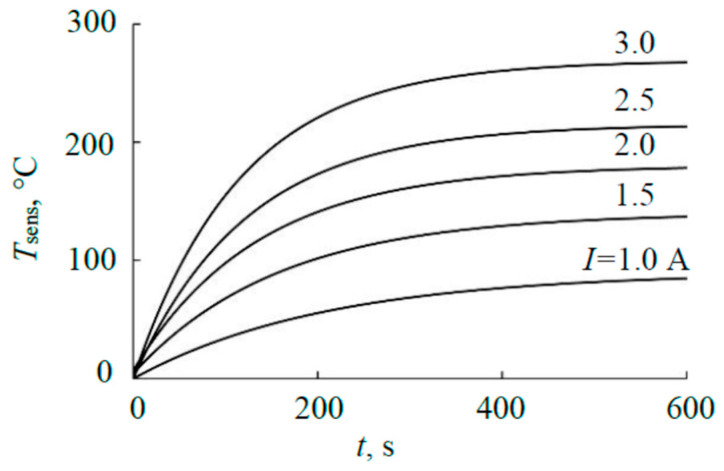
Kinetic heating curves of the sensor sensing element at different discharge currents.

**Figure 15 materials-17-03599-f015:**
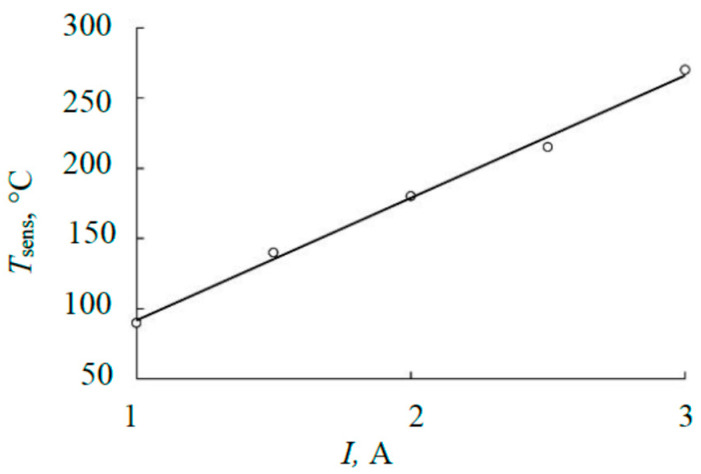
Temperature of the sensor sensing element in the steady state.

**Figure 16 materials-17-03599-f016:**
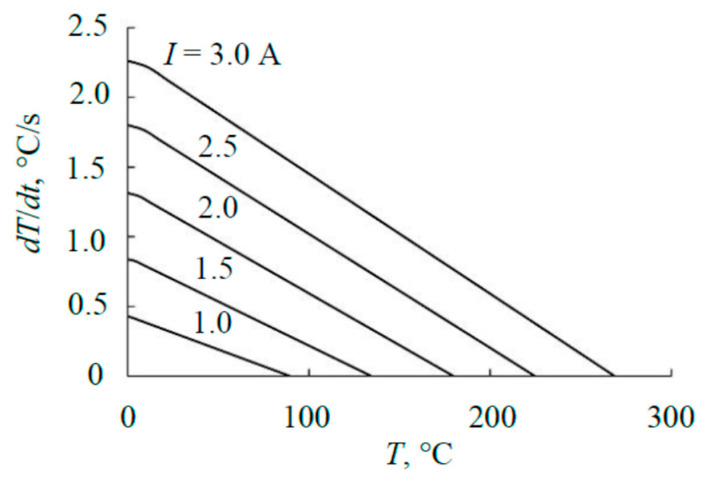
Heating kinetics of the sensing element.

**Figure 17 materials-17-03599-f017:**
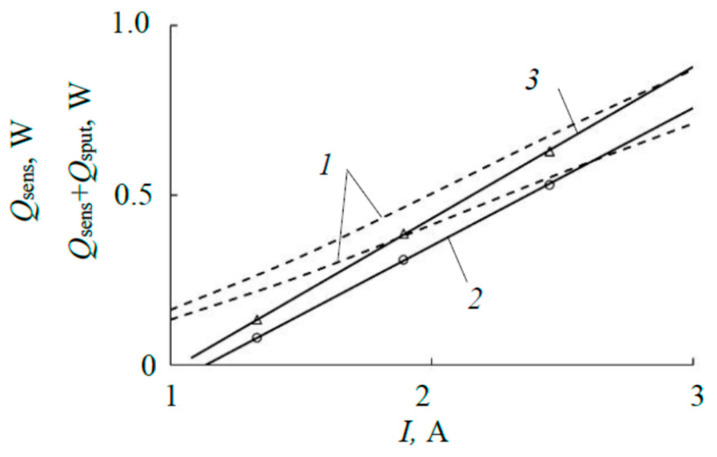
Heat flow absorbed by the sensing element: *1*—area of experimental values with measurement error ± 10%; *2*—modeling result (*Q_sens_*); *3*—modeling result with additional phenomena taken into account (*Q_sens_* + *Q*_sput_).

**Table 1 materials-17-03599-t001:** Viewing angles for some fragments of the external plate.

*i*	0	10	20	30	40	50	60	65
φ*_i_*, 10^−3^	2.62	2.59	2.52	2.42	2.42	2.15	1.99	1.93

**Table 2 materials-17-03599-t002:** Values of parameter *A*.

*I*, A	1.0	1.5	2.0	2.5	3.0
*A*, °C/s	0.43	0.88	1.34	1.76	2.31
*Q_sens_*, W	0.15	0.30	0.46	0.60	0.79

**Table 3 materials-17-03599-t003:** Calculation results using Formula (12).

*I*, A	1.0	1.5	2.0	2.5	3.0
*Q*_sput_, W	0.04	0.06	0.08	0.10	0.12

## Data Availability

Data are contained within the article.

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
