# Peer review of "Study of Heat Flow at Substrate during Sputtering of Copper–Titanium Sandwich Target"

_materials, 2024, doi:10.3390/ma17143599_

Round 1

Reviewer 1 Report

Comments and Suggestions for Authors

In this paper, the heat flow generated by sputtering of the sandwich target of Cu-Ti sandwich structure in argon was studied using the heat transfer module of COMSOL Multiphysics. The two-dimensional homogeneous Fourier equation was numerically solved to determine the surface temperature distribution and heating dynamics of the outer plate of the concentric radiation source, and the calculated data were compared with the experimental data. This study has certain significance, but it has the following deficiencies, which need to be modified:

1. The title of this paper does not match the content in the paper, so it is suggested to use a title that is closer to the research content;

2. The abstract of this paper lacks some quantifiable computational data and experimental data, not to mention some qualitative or quantitative conclusions, not to mention deep-level innovative scientific conclusions, so it is difficult to arouse readers' interest in this paper, please rewrite it;

3. In the Introduction part, the status quo and shortcomings of relevant research related to this paper are not stated, which is very simple. I hope the author will carefully and deeply search for relevant literature for elaboration;

4. In the last paragraph of the Introduction, the author still makes an introduction by reference, but does not summarize the differences between this paper and previous literature studies, so as to reflect the value of this study, please revise;

5. The format of Table 1, Table 2, and Table 3 in the paper is not standard, please modify it according to the standard three-line table;

6. In the results and discussion part of the paper, there is no actual cloud map of numerical simulation analysis, so I hope the author can add it to increase credibility;

7. The conclusion of this paper is too tedious and needs to be simplified. It is suggested to rewrite it to highlight the innovation of this paper.

Reviewer 2 Report

Comments and Suggestions for Authors

The authors demonstrated the study of heat flow generating by a magnetron sandwich target sputtered in argon. Further using the COMSOL Multiphysics model was used to study the distribution of the external plate surface temperature and its heating kinetics are determined by numerical solution of a two-dimensional homogeneous Fourier equation using the Heat Transfer Module. The results were satisfactory and the research topic is nice. However, there are some minute errors that authors need to be rectified before the final acceptance in Materials journal.

1.      Novelty of your work needs to be enriched more. In the introduction please specify the significance of the application of coating of films.  

2.      Please check the errors in the image’s formation (edit the pictures edges) and improve the quality of the images.

3.      It could be better if authors should expand the u, v, P, W, t, T, K like current, power, etc., and their units for the better understanding toe the readers.

4.      While the magnetron sputtering what kind of gas was used to excite the plasmonic source and it alos varies the partial pressure inside chamber with the flow rate and ratio.

5.      There are some typo errors, grammatical errors and prefix/suffix errors in equation/formulas in the manuscript. Please check the entire manuscript carefully and do the quick proof read.

6.      Conclusion part should be trimmed, please consider it.

7.      Some of the references are not upto the mark, please include the latest published references to further enhance your novelty. Consider the following reference in the introduction part Applied Nano 2 (1), 46-66.

Reviewer 3 Report

Comments and Suggestions for Authors

This manuscript deal with Study of Thermal Processes During Sputtering of Copper-Titanium Sandwich Target. It is very interesting for potential readers and researchers.

For Next step process, authors respond to questions and comments. 

1. In the introduction, I recommend you would show originality and motivation compared to other researchers' work 

2. in conclusion, you would summary more shortly. 

Comments on the Quality of English Language

Please check typo and missing in the manuscript. 
